# A novel Visible Multilayer Concept Factorization for Image Data Representation and Clustering

## Abstract

Traditional Concept Factorization (CF) methods learn feature of one data point from high-dimensional data space in the form of vector, which leads to the loss of pixel-level neighborhood information in Two-Dimensional (2D) images. In light of this, we present a novel Visible Multilayer Concept Factorization for image-data representation, termed VMCF. Specifically, to uncover deep latent features from complex data, VMCF adopts a multilayer framework, equipped with a 'Decomposition, Dimensionality reduction and Data reconstruction' network ($D^3$-net) in each layer. To obtain locality-preserving features, $D^3$-net firstly performs adaptive graph regularized concept learning on the input data of each layer. Then, $D^3$-net performs 2D feature extraction over the obtained basis images in order to reduce the loss of pixel-level neighborhood information during dimension plunging. The reconstructed data formed by the improved basis images and coefficient matrix is used as input for the next layer. In this way, the dimensions of the original data can gradually decrease at each layer, avoiding information loss caused by sudden dimensionality reduction. Meanwhile, 2D-reduced basis images can mediately improve the quality of new data representations. Extensive numerical experiments on several public image databases have shown that VMCF outperforms other state-of-the-art algorithms.

## 1 Introduction

The number and dimensionality of image data in real-world applications have shown explosive growth with the popularization of high-definition devices, posing challenges for image data processing and analysis. Luckily, Representation Learning (RL) techniques can well handle them by extracting latent features, from data with high-dimensionality (Bengio et al., 2013). Therefore, RL has become a trending topic in fields of image processing, data mining and computer vision.

In this paper, we focus on matrix-based RL algorithms since image data is often stored and calculated in the form of matrix composed of pixel values in practical applications. Matrix Factorization (MF) has been proven to be powerful for feature learning and clustering (Wang & Zhang, 2012), including classical methods like Principle Component Analysis (PCA) (Wold et al., 1987), Nonnegative Matrix Factorization (NMF) (Lee & Seung, 1999) and Concept Factorization (CF) (Xu & Gong, 2004). Note that, benefiting from nonnegative constraints, NMF and CF are more suitable for processing image data since image pixel values are typically nonnegative data. NMF aims to decompose a nonnegative data matrix $X$ into two matrices, whose product is the approximation to $X$. One factor indicates basis vectors and the other factor corresponds to the parts-based representations. Benefit by non-negative constraints, the basis vectors obtained from NMF are also non-negative and can be considered as part of the original images. Taking facial image as an example, the obtained basis vectors may be local parts of the face such as the nose, eyes, cheeks, etc.

On the basis of retaining the non-negative constraints, CF differs from NMF in that it can also be performed in the Reproducing Kernel Hilbert Space by further decomposing the base matrix into $X$ itself and an auxiliary coefficient matrix. With both nonlinearity and high interpretability, CF yields unusually brilliant results in data dimensionality reduction and clustering tasks (Li et al.,

2024). Subsequently, an influx of CF variants have been introduced to cater to diverse application scenarios, for example, robust ones (Yang et al., 2022) and multiview ones (Mu et al., 2023). One of the most common branches is locality-preserving CF (Chen & Li, 2020). In the process of concept factorization, keeping locality manifold information of data points can make sure their relative positional relationships (or similarity relationships) in high-dimensional data space can be retained even in the mapped low dimensional space. That could potentially improve the discriminability of features, thereby enhancing their performance in subsequent high-level tasks, such as data classification and clustering (Zhang et al., 2021). To achieve this goal, various locality-preserving strategies have been introduced into traditional CF model, which can be roughly divided into three categories, that is, graph-regularized ones, local coordinate coding driven ones and self-representation based ones. Graph-regularized CF methods, represented by Locally Consistent CF (LCCF) (Cai et al., 2010), typically incorporate the graph Laplacian regularization into CF. Local coordinate coding driven CF mainly incorporate the principle of local coordinate coding (LCC) into traditional CF to retain local information in data. Illustrative methods in this group include Local Coordinate CF (LCF) (Liu et al., 2013), and Graph-based Local concept coordinate factorization (GLCF) (Li et al., 2015a). Different from above two kinds of CF methods, self-representation based ones regard the input data as dictionary and then construct the affinity matrix using the new representation (i.e., coefficient matrix) rather than the original raw data. Representative methods include SRMCF (Ma et al., 2018) and JSGCF (Peng et al., 2019).

However, when dealing with image data, these traditional local preservation CF methods generally face several challenges. First, the local manifold structure between samples cannot be adaptively maintained during concept factorization iterations. Second, existing CF methods represent an image as a one-dimensional vector, and then concatenate a set of vectors into a matrix, which serves as the input for CF. Such approach not only results in loss of pixel-level structural information within the image, but also leads to higher computational complexity due to the large size of the matrix. Third, direct reduction of higher image dimensions to a very small value through a one-step calculation may also lead to the risk of important valuable information loss.

In view of this, we propose a novel framework to inherit the advantages of existing CF models and meanwhile overcome above-mentioned drawbacks. Main contributions of this work are summarized as follows:

(1) Technically, a Visible Multilayer Concept Factorization (VMCF), is proposed for image data representation and clustering. Aimed at improving representation power, VMCF is equipped with a 'Decomposition-Dimensionality reduction-Data reconstruction' network (shortly, $D^3$-net) in each layer. For the input data matrix of each layer, $D^3$-net decomposes them by adaptive graph regularized CF into two nonnegative parts, i.e., set of basis images and coefficient matrix. Then, $D^3$-net extracts two-dimensional features from the basis images and finally reconstruct the input data based on this.

(2) To achieve local-preserving representations of image data, the innovations of VMCF are twofold. a) preserving sample-level local information by introducing an adaptive adjacency matrix into CF framework to record and update neighborhood relationships between samples; b) preserving pixel-level local information for basis images by restoring them to a set of two-dimensional matrices and perform 2D feature extraction on it. Obtained basis images with higher quality and lower dimensions can not only reduce the computational complexity of the model, but also indirectly improve the accuracy of the coordinate matrix.

(3) To uncover hidden deep features, VMCF adopts a multi-layer structure. The multilayer structure can not only excavate deeper features than traditional shallow structures, but also reduce the dimensionality of raw data layer by layer to avoid the loss of valuable information caused by too fast decrease in data dimensionality.

The paper is outlined as follows. In Section 2, we briefly review related works. In Section 3, we present VMCF in detail. We show the optimization procedures in Section 4. Section 5 describes the simulation results. The paper is finally concluded in Section 6.

## 2 RELATED WORKS

In this section, we briefly review related algorithms, i.e., Concept Factorization (CF) (Xu & Gong, 2004) and Two-Dimensional Principle Component Analysis (2DPCA) (Yang et al., 2004).

## 2.1 CONCEPT FACTORIZATION (CF)

Given a nonnegative data matrix $X = [x_1, x_2, \ldots, x_N] \in \mathbb{R}^{D \times N}$ where $D$ is the original dimensionality of $N$ samples, Concept Factorization mainly aims to calculate two nonnegative matrices $W \in \mathbb{R}^{N \times r}$ and $V \in \mathbb{R}^{r \times N}$, to make the product of $XWV$ approximates to $X$ itself. That is, the task of CF is to solve the following mathematical problem:

$$\min_{W,V} J = \|X - XWV\|_F^2, \, s.t. W, V \geq 0 \tag{1}$$

In the principle of CF, $XW$ and $V$ can be regarded as $r$ bases and corresponding $N$ coordinates (or weights) respectively.

## 2.2 TWO-DIMENSIONAL PRINCIPLE ANALYSIS (2DPCA)

Given an image matrix $A \in \mathbb{R}^{m \times n}$, 2DPCA tries to learn features $Y = AX$ by projecting $A$ onto a $n$-dimensional projection vector $X$. Ideally, an optimal projection $X$ should maximize the total scatter of the projected features $Y$. Thus, 2DPCA gives the following criterion.

$$\arg\max_X J(X) = trace(S_x) \tag{2}$$

where $S_x$ denotes the covariance matrix of the projected features $Y$, and $trace(S_x)$ denotes the trace of the matrix $S_x$. The definition of $S_x$ and derivation process are as follows

$$\begin{aligned} S_x &= \mathrm{E}(Y - \mathrm{E}Y)^T \\ &= \mathrm{E}\left[AX - \mathrm{E}(AX)\right]\left[AX - \mathrm{E}(AX)\right]^T \\ &= \mathrm{E}\left[(A - \mathrm{E}A)X\right]\left[(A - \mathrm{E}A)X\right]^T \end{aligned} \tag{3}$$

According to $trace(AB) = trace(BA)$, Eq.(2) can be rewritten as

$$\arg\max_X J(X) = X^T \mathrm{E}\left[(A - \mathrm{E}A)^T (A - \mathrm{E}A)\right] X \tag{4}$$

Let $G_t = \mathrm{E}\left[(A - \mathrm{E}A)^T (A - \mathrm{E}A)\right]$, we can easily find $G_t$ is the image covariance matrix of matrix $A$, which can be directly obtained by the training samples. Suppose that there are $M$ images in total, $G_t$ can be evaluated by

$$G_t = \frac{1}{M} \sum_{j=1}^{M} (A_j - \bar{A})^T (A_j - \bar{A}) \tag{5}$$

Since we usually need to have more than one optimal projection axis $X_{opt}$, the solutions of Eq.(4) can be composed by $X_{opt} = [X_1, \ldots, X_d]$, where $X_1, \ldots, X_d$ are orthonormal eigenvectors of $G_t$ corresponding to the $d$ largest eigenvalues.

# 3 VISIBLE MULTILAYER CONCEPT FACTORIZATION (VMCF)

## 3.1 FRAMEWORK OF VMCF

To uncover deep hidden features of image data, VMCF designs a novel multilayer structure, equipped with a D$^3$-net in each layer, that is, 'Dcomposition - Dimensionality reduction - Data reconstruction' network. Specifically, we represent the input image data matrix for each layer as $X^{(m-1)} = \left[x_1^{m-1}, x_2^{m-1}, \ldots, x_N^{m-1}\right] \in \mathbb{R}^{(w_{m-1} \times h_{m-1}) \times N}$, with its column is the 1D representation of one picture, $m = 1, 2, \ldots, M$ is the number of layer, and $N$ is the number of samples. Note that, $w$ and $h$ denote width and height of an image and $w_{m-1} \times h_{m-1}$ is the dimensionality of $X^{(m-1)}$, and especially, $w_0 \times h_0$ is the dimensionality of input data $X^{(0)}$ of VMCF. D$^3$-net first decomposes $X^{(m-1)}$ into two parts, that is, basis images $X^{(m-1)}W^{(m-1)} \in \mathbb{R}^{(w_{m-1} \times h_{m-1}) \times r}$ and coefficients $V^{(m-1)} \in \mathbb{R}^{r \times N}$, using the Adaptive graph regularized CF, where $r$ is the rank. Then, D$^3$-net unfold each basis image into 2D representation and reduce its dimensionality to $w_m \times h_m$ in both row and column directions using 2DPCA. In this way, the dimensionality of the image

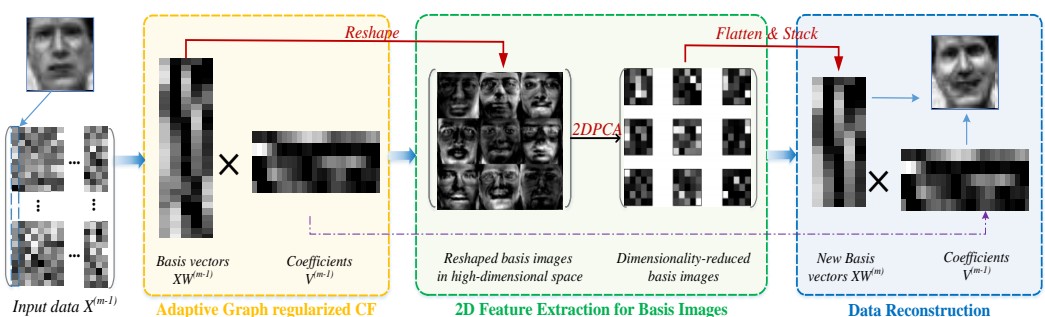

Figure 1: The D$^3$-net flowchart, illustrating a single layer of our proposed VMCF

data has slowly decreased, and the neighborhood relationships between image pixels have also been retained. The obtained improved low dimensional basis images may also potentially enhance the quality of subsequent reconstructed data. D$^3$-net afterwards flatten the newly obtained basis images to $(XW)_{new}^{(m)} \in \mathbb{R}^{(w_m \times h_m) \times r}$, and reconstruct $\hat{X}^{(m)}$ by $(XW)_{new}^{(m)} V^{(m-1)}$. Reconstructed $\hat{X}^{(m)}$ can proceed to the next layer after being detected by the error analysis module. The framework of D$^3$-net is shown in Figure 1.

## 3.2 ADAPTIVE GRAPH REGULARIZED CF

In this subsection, we introduce the first module of D$^3$-net: 'Decomposition', in detail. To decompose the input data and to learn locality-preserving new representations adaptively, we define a novel Adaptive graph regularized Concept Factorization, termed ACF. For the input matrix $X$, ACF aims to find two nonnegative matrices $W \in \mathbb{R}^{N \times r}$ and $V \in \mathbb{R}^{r \times N}$, whose product $XWV$ is approximate to $X$ itself. $XW$ can be seen as a set of basis vectors, or as basis images in the context of image data, and $V$ is its corresponding coordinate matrix. To enable matrix $V$ have the ability to retain local manifold information automatically without specifying the neighborhood parameters, ACF introduces the adaptive graph regularization term into root CF. Specifically, to automatically keep the local manifold information of feature space, an adaptive weight matrix $Q = [q_1, q_2, \ldots, q_N] \in \mathbb{R}^{N \times N}$ is introduced to minimize the neighborhood reconstruction errors over $V$, that is, $\sum_{i=1}^{N} \|V - Vq_i\|_F^2$. By combining the adaptive graph regularization term and the standard concept factorization term, we can obtain the objective function of ACF in the form of matrix as follows.

$$\min_{W,V,Q} \|X - XWV\|_F^2 + \alpha \|V - VQ\|_F^2 \\ s.t. W, V, Q \geq 0, Q_{ii} = 0, i = 1, 2, \ldots, N \tag{6}$$

where $W, V, Q \geq 0$ are the nonnegative constraints, $\alpha \geq 0$ is trade-off parameter, and $Q_{ii} = 0$ is added to avoid the trivial solution $Q = I$. Substituting Eq.(6) into the scenario of the $m$-th layer of VMCF, we can obtain the following formula.

$$\min_{W^{(m-1)}, V^{(m-1)}, Q^{(m)}} \left\|X^{(m-1)} - X^{(m-1)}W^{(m-1)}V^{(m-1)}\right\|_F^2 + \alpha \left\|V^{(m-1)} - V^{(m-1)}Q^{(m)}\right\|_F^2 \\ s.t. W^{(m-1)}, V^{(m-1)}, Q^{(m)} \geq 0, Q_{ii}^{(m)} = 0, i = 1, 2, \ldots, N \tag{7}$$

## 3.3 2D FEATURE EXTRACTION AND DATA RECONSTRUCTION

Basis matrix $XW$ obtained by ACF conveys dual message. In one respect, for image data, it can be seen as basis images, which means that taking facial data as an example, the visualized basis matrix is the partial representation of the full face, such as nose, mouth, cheeks, etc. In another respect, since CF series methods iteratively update $XW$ and $V$, $XW$ potentially affect the quality of the reconstructed data $X$ and the new representation $V$. Therefore, in 'Dimensionality reduction' module in D$^3$-net, we reduce the dimensionality and preserve pixel-level locality in basis images. We first reshape $(XW)^{(m-1)}$ obtained in the $m$-th layer into $r$ individual $w_{m-1}$ by $h_{m-1}$ matrices,

where each matrix $A_i$, $i = 1, 2, \ldots, r$, represents one basis image. D$^3$-net then performs 2D feature extraction over each basis image by 2DPCA (Yang et al., 2004). Note that 2DPCA can only work in the column direction of images, to project basis images onto a low-dimensional feature space in both row and column directions, D$^3$-net performs two-directional 2DPCA over $A_i$. Suppose that the dimensionality of each basis image is reduced from $w_{m-1} \times h_{m-1}$ to $w_m \times h_m$, where $w_m < w_{m-1}$, $h_m < h_{m-1}$, and two projection matrices for dual directions are $C \in \mathbb{R}^{w_{m-1} \times w_m}$ and $R \in \mathbb{R}^{h_{m-1} \times h_m}$, which reflects information between columns and rows of images respectively. Basis image $A_i$ is projected onto $C$ and $R$ simultaneously, yielding a $w_m$ by $h_m$ matrix $Y_i = R^T A_i C$. $Y_i$ is the low-dimensional feature matrix and can be used to reconstruct the basis image $A_i$ by $\hat{A}_i = R Y_i C^T$. D$^3$-net finally obtains new basis matrix $(XW)_{new}^{(m)}$ by flattening each $Y_i$ to a $(w_m \times h_m) \times 1$ vector and concatenating $r$ vectors into a matrix. Subsequently, the reconstructed $\hat{X}^{(m)}$ in the $m$-th layer can be obtained by calculating $(XW)_{new}^{(m)} V^{(m-1)}$.

## 4 OPTIMIZATION OF ACF

In this section, we show how to optimize the objective function of ACF in the 'Decomposition' module in VMCF. For the sake of simplicity, we first discuss the general form of the objective function in Eq.(6). We can find that the involved variables, i.e., $W$, $V$, and $Q$ depend on each other, so they cannot be solved directly. As common procedures (Zhao & Tan, 2017), we solve it by using the Multiplicative Update Rules (MUR) to obtain local optimal solutions. To be specific, we solve it by updating the variables alternately and compute one of the variables each time by fixing others.

Firstly, we rewrite the objective function in the form of matrix traces as follows.

$$\min J(W, V, Q) = tr\left((X - XWV)^T (X - XWV)\right) + \alpha tr\left((V - VQ)^T (V - VQ)\right) \tag{8}$$
$$s.t. W, V, Q \geq 0, Q_{ii} = 0, i = 1, 2, \ldots, N$$

where $tr(\cdot)$ denotes the trace of matrix.

**1) Fix others and update the factors $W$ and $V$.** We first show how to optimize $W$ and $V$. Let $\psi_{ik}$ and $\phi_{ik}$ be the Lagrange multipliers for the constraints $w_{ik} \geq 0$, $v_{ik} \geq 0$, and $\Psi = [\psi_{ik}]$, $\Phi = [\phi_{ik}]$, the Lagrange function $\mathcal{L}_1$ for $W$ and $V$ can be reconstructed as follows.

$$\mathcal{L}_1 = tr\left((X - XWV)^T (X - XWV)\right) + \alpha tr\left(VLV^T\right) + tr\left(\Psi W^T\right) + tr\left(\Phi V^T\right) \tag{9}$$

where $L = (I - Q)(I - Q)^T$, $I$ is an identity matrix. Then, $W$ and $V$ can be alternately updated by fixing others. The derivatives w.r.t. $W$ and $V$ can be formulated as

$$\partial \mathcal{L}_1 / \partial W = 2X^T XWVV^T - 2X^T XV^T + \Psi \tag{10}$$

$$\partial \mathcal{L}_1 / \partial V = 2W^T X^T XWV - 2W^T X^T X + 2\alpha VL + \Phi \tag{11}$$

By applying the KKT conditions (Wu, 2007) $\psi_{ik} w_{ik} = 0$ and $\phi_{ik} v_{ik} = 0$, we can obtain the following equations for $\psi_{ik}$ and $\phi_{ik}$:

$$\left(X^T XWVV^T\right)_{ik} w_{ik} - \left(X^T XV^T\right)_{ik} w_{ik} = 0 \tag{12}$$

$$\left(X^T XWVV^T\right)_{ik} v_{ik} - \left(X^T XV^T\right)_{ik} v_{ik} + \alpha(VL)_{ik} v_{ik} = 0 \tag{13}$$

Note that the above two equations can lead to the following multiplicative updating rules for the basis images $W$ and the corresponding coefficients $V$.

$$w_{ik} \leftarrow w_{ik} \frac{\left(X^T XV^T\right)_{ik}}{\left(X^T XWVV^T\right)_{ik}} \tag{14}$$

$$v_{ik} \leftarrow v_{ik} \frac{\left(X^T XV^T\right)_{ik}}{\left(X^T XWVV^T + \alpha VL\right)_{ik}} \tag{15}$$

**2) Fix others and update the adaptive weights $Q$.** With the computed $W$ and $V$, the adaptive weights $Q$ also can be updated by solving the formulation in Eq.(8) in a similar way as $W$ and $V$. Since $W$ and $V$ can be seen as constants in this step, we can reformulate Eq.(8) as

$$\min J(Q) = tr\left((V - VQ)^T (V - VQ)\right) \tag{16}$$
$$s.t. Q \geq 0, Q_{ii} = 0, i = 1, 2, \ldots, N$$

Table 1: List of Used Datasets and Dataset Information

| Database | Data type | #Points | #Dim | #Class |
|----------|-----------|---------|------|--------|
| AR | Face images | 2600 | 32×32 | 100 |
| MIT CBCL | Face images | 3240 | 32×32 | 10 |
| CMU PIE | Face images | 11554 | 32×32 | 68 |
| ETH80 | Object images | 3280 | 32×32 | 80 |

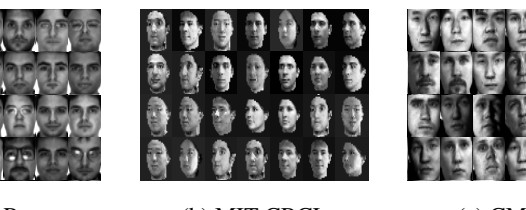 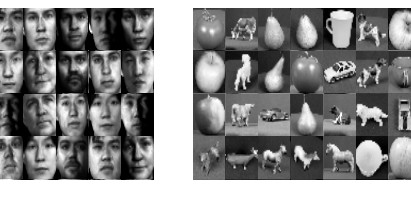

     (a) AR            (b) MIT CBCL         (c) CMU PIE         (d) ETH80

Figure 2: Examples of evaluated databases

Let $\tau_{ik}$ be a Lagrange multiplier for the constraint $q_{ik} \geq 0$ and $\Gamma = [\tau_{ik}]$, the Lagrange function for $Q$ can be defined as

$$\mathcal{L}_2 = tr\left((V - VQ)^T (V - VQ)\right) + tr\left(\Gamma Q^T\right) \tag{17}$$

Then we can compute the derivative of $\mathcal{L}_\in$ w.r.t. $Q$ as

$$\partial \mathcal{L}_2 / \partial Q = 2\left(V^T V Q - V^T V\right) + \Gamma \tag{18}$$

Using the KKT condition $\tau_{ik} q_{ik} = 0$ we can obtain:

$$\left(V^T V Q - V^T V\right)_{ik} q_{ik} = 0 \tag{19}$$

which leads to the following updating rule for weights $Q$:

$$q_{ik} \leftarrow q_{ik} \frac{\left(V^T V\right)_{ik}}{\left(V^T V Q\right)_{ik}}, \text{and } q_{ii} = 0 \tag{20}$$

After $Q$ is obtained in each iteration, we can return it to further update matrices $W$ and $V$.

## 5 SIMULATION RESULTS AND ANALYSIS

In this section, we conduct simulations to examine our proposed VMCF for data representation and clustering. Four public image databases are evaluated, including three commonly-used face databases AR (Martinez & Benavente, 1998), MIT CBCL (Rowley et al., 1998), CMU PIE (Sim et al., 2002) and one object database ETH80 (Leibe & Schiele, 2003). Detailed information of these used databases is shown in Table.1. Some samples are shown in Figure 2. The experimental results of our method are compared with those of seven related algorithms, i.e., shallow models including CF (Xu & Gong, 2004), LCF (Liu et al., 2013), SRMCF (Ma et al., 2018), LGCF (Li et al., 2017a) and deep models including MCF (Li et al., 2015b), GMCF (Li et al., 2017b), and DSCF-net (Zhang et al., 2020). We perform all the simulations on a PC with Intel Core i7-10700 CPU @ 2.90 GHz.

### 5.1 EXPERIMENTAL SETUP

**1) Clustering Evaluation Process.** Following the common procedures in (Cai et al., 2010) and (Yang et al., 2024), we perform $K$-means algorithm with cosine distance on the obtained new representation from each evaluated model. For each number $K$ of clusters, we randomly choose $K$ categories from each database and use the $K$ categories data to form the matrix $X$. The rank $r$ of

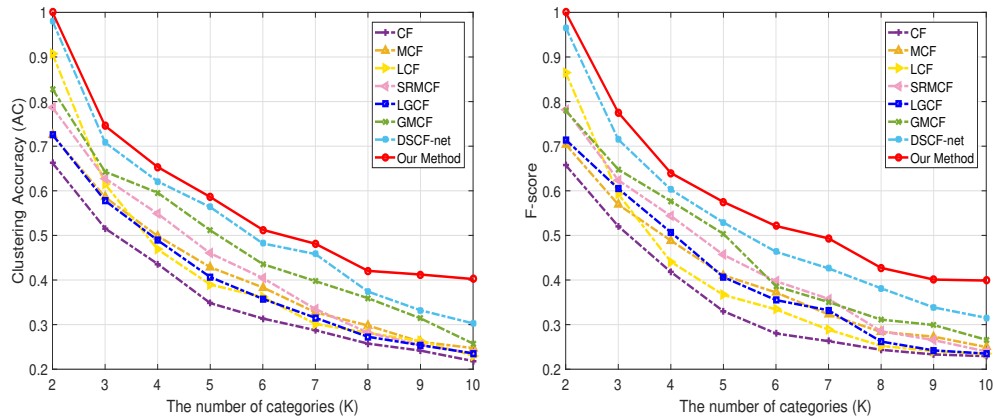

Figure 3: Clustering performance on AR database

matrix is set to $K + 1$ for all the evaluated methods according to (Sugiyama, 2007). For fair comparison, we average the numerical results over 20 random initializations for the $K$-means algorithm.

**2) Evaluation Metrics.** Two widely-used quantitative evaluation metrics are used in our simulations, i.e., Clustering Accuracy (AC) and $F$-score. AC is the percentage of the computed cluster labels to the true labels provided by the original databases. AC can be obtained by

$$AC = \sum_{i=1}^{N} \delta\left(r_i, map\left(p_i\right)\right)/N \tag{21}$$

where $N$ is the number of samples and the function $map\left(p_i\right)$ is the permutation mapping function that maps the cluster label $p_i$ obtained by the clustering method to the true label $r_i$ provide by the original data corpus.

$F$-score is also a commonly used clustering quantification indicator, which is defined as

$$F_\mu = \frac{\left(\mu^2 + 1\right) PRECISION \times RECALL}{\mu^2 PRECISION + RECALL} \tag{22}$$

where we set the parameter $\mu = 1$ in our simulations.

**3) Parameter Selection for Multilayer Models.** Note that there are four multilayer models tested in our simulations, including MCF, GMCF, DSCF-net and our VMCF. According to previous works in (Zhang et al., 2021) and (Zhang et al., 2020), we set the number of model layers to 3 for all the evaluated multilayer CF models. And we specify the target dimensions of basis images, i.e., $w_m$ and $h_m$, $m = \{1, 2, 3\}$ in these 3 layers of our VMCF as $\{24, 24\}$, $\{16, 16\}$, and $\{8, 8\}$.

### 5.2 QUANTITATIVE CLUSTERING RESULTS AND ANALYSIS

**1) Face Image Clustering.** We firstly test representation and clustering power of each method on three face image databases. Clustering performance in terms of AC and $F$-score over varied values of $K$ is tested. The value of $K$ ranges from 2 to 10 with step 1. The clustering curves of AC and $F$-score are reported in Figures 3-5. The averaged AC and $F$-score according to the curves are summarized in Tables 2-4. We can easily find that: (1) the increasing value of $K$ can decrease the AC and $F$-score of each model since the clustering data of less categories is relatively easier; (2) multilayer models roughly achieve better clustering results than single-layer ones, which indicates that multilayer structure can indeed improve representation and clustering performance than shallow structure; (3) our VMCF delivers higher AC and $F$-score than the other baselines in most cases.

**2) Object Image Clustering.** Then we evaluate each model for representing and clustering the object image data over ETH80 database. The clustering curves are shown in Figure 6, and the averaged values of ACs and $F$-scores are reported in Table 5. We can see that the clustering curves also show a downward trend with the increase of the number of categories. And our proposed VMCF outperforms the other baseline methods in this study.

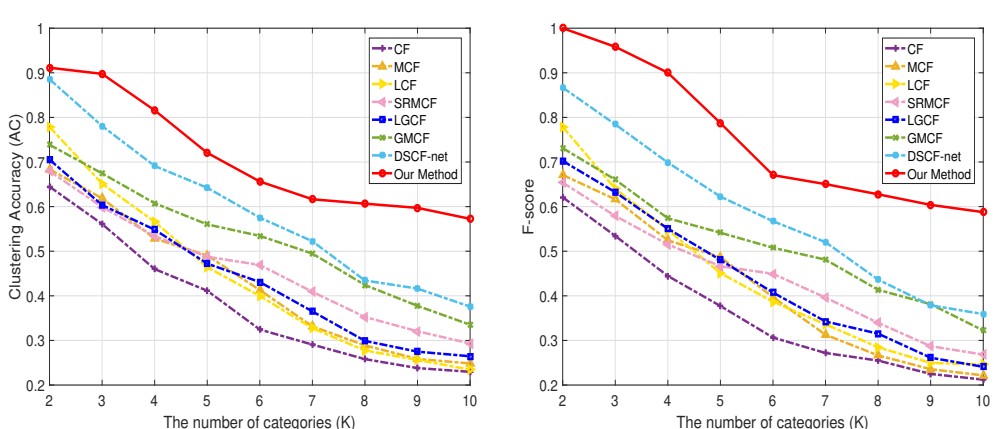

Figure 4: Clustering performance on MIT CBCL database

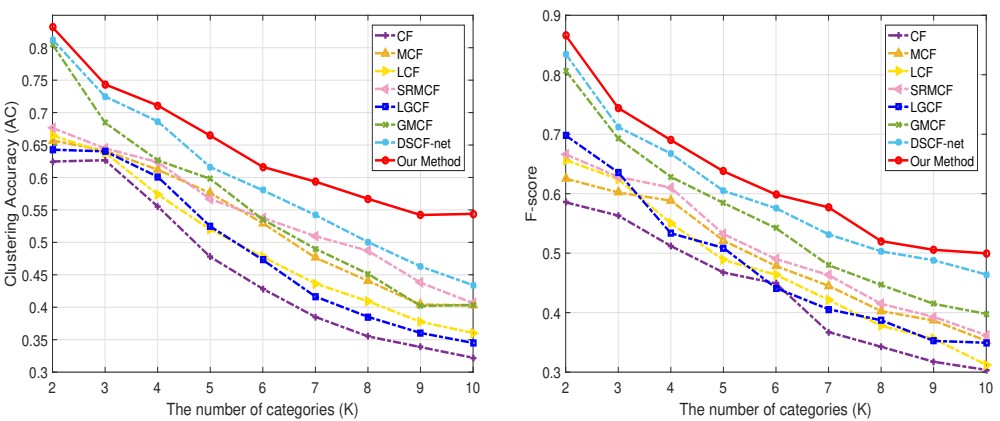

Figure 5: Clustering performance on CMU PIE database

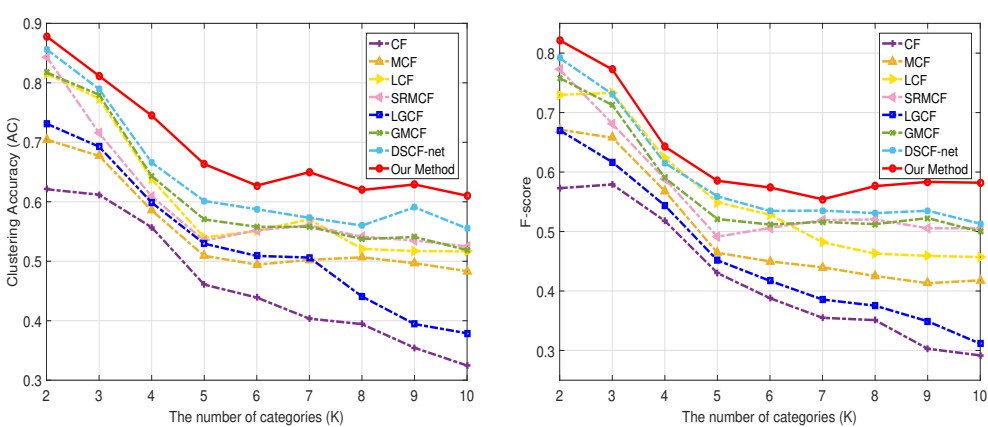

Figure 6: Clustering performance on ETH80 database

| Method | AC | | | $F$-score | | |
|---|---|---|---|---|---|---|
| | Mean | std | Best | Mean | std | Best |
| CF | 0.3645 | 0.1472 | 0.6623 | 0.3528 | 0.1503 | 0.6575 |
| MCF | 0.4173 | 0.1607 | 0.7252 | 0.4082 | 0.1530 | 0.7038 |
| LCF | 0.4243 | 0.2166 | 0.9067 | 0.4017 | 0.2083 | 0.8648 |
| SRMCF | 0.4372 | 0.1873 | 0.7865 | 0.4390 | 0.1819 | 0.7818 |
| LGCF | 0.4038 | 0.1655 | 0.7257 | 0.4066 | 0.1695 | 0.7145 |
| GMCF | 0.4824 | 0.1811 | 0.8279 | 0.4579 | 0.1790 | 0.7799 |
| DSCF-net | 0.5358 | 0.2139 | 0.9802 | 0.5264 | 0.2090 | 0.9660 |
| **Ours** | **0.5793** | **0.1964** | **1.0000** | **0.5813** | **0.1992** | **1.0000** |

Table 2: Averaged AC and $F$-score values of evaluated methods based on AR database

| Method | AC | | | $F$-score | | |
|---|---|---|---|---|---|---|
| | Mean | std | Best | Mean | std | Best |
| CF | 0.3798 | 0.1495 | 0.6447 | 0.3608 | 0.1448 | 0.6209 |
| MCF | 0.4291 | 0.1603 | 0.6837 | 0.4146 | 0.1679 | 0.6710 |
| LCF | 0.4395 | 0.1912 | 0.7787 | 0.4357 | 0.1865 | 0.7789 |
| SRMCF | 0.4605 | 0.1300 | 0.6794 | 0.4394 | 0.1306 | 0.6538 |
| LGCF | 0.4404 | 0.1555 | 0.7052 | 0.4370 | 0.1648 | 0.7024 |
| GMCF | 0.5274 | 0.1346 | 0.7387 | 0.5125 | 0.1316 | 0.7311 |
| DSCF-net | 0.5915 | 0.1738 | 0.8856 | 0.5816 | 0.1781 | 0.8669 |
| **Ours** | **0.7105** | **0.1327** | **0,9114** | **0.6975** | **0.1370** | **0.9223** |

Table 3: Averaged AC and $F$-score values of evaluated methods based on MIT CBCL database

| Method | AC | | | $F$-score | | |
|---|---|---|---|---|---|---|
| | Mean | std | Best | Mean | std | Best |
| CF | 0.4571 | 0.1202 | 0.6266 | 0.4343 | 0.1063 | 0.5855 |
| MCF | 0.5267 | 0.1001 | 0.6571 | 0.4892 | 0.1005 | 0.6253 |
| LCF | 0.4953 | 0.1113 | 0.6646 | 0.4724 | 0.1195 | 0.6565 |
| SRMCF | 0.5433 | 0.093 | 0.6763 | 0.5067 | 0.1096 | 0.6663 |
| LGCF | 0.4877 | 0.1193 | 0.6428 | 0.4791 | 0.1247 | 0.6979 |
| GMCF | 0.5551 | 0.1359 | 0.8048 | 0.5548 | 0.1371 | 0.8068 |
| DSCF-net | 0.5955 | 0.1266 | 0.8125 | 0.5980 | 0.1214 | 0.8348 |
| **Ours** | **0.6460** | **0.0997** | **0.8316** | **0.6266** | **0.1228** | **0.8661** |

Table 4: Averaged AC and $F$-score values of evaluated methods based on CMU PIE database

| Method | AC | | | $F$-score | | |
|---|---|---|---|---|---|---|
| | Mean | std | Best | Mean | std | Best |
| CF | 0.4630 | 0.1094 | 0.6211 | 0.4209 | 0.1110 | 0.5791 |
| MCF | 0.5511 | 0.0848 | 0.7046 | 0.5008 | 0.1038 | 0.6713 |
| LCF | 0.6048 | 0.1144 | 0.8152 | 0.5583 | 0.1120 | 0.7330 |
| SRMCF | 0.6019 | 0.1086 | 0.8439 | 0.5655 | 0.0983 | 0.7725 |
| LGCF | 0.5313 | 0.1234 | 0.7314 | 0.4579 | 0.1249 | 0.6702 |
| GMCF | 0.6137 | 0.1108 | 0.8173 | 0.5716 | 0.0970 | 0.7581 |
| DSCF-net | 0.6422 | 0.1087 | 0.8565 | 0.5940 | 0.1003 | 0.7917 |
| **Ours** | **0.6927** | **0.0965** | **0.8782** | **0.6325** | **0.0970** | **0.8215** |

Table 5: Averaged AC and $F$-score values of evaluated methods based on ETH80 database

## 5.3 RUNTIME COMPARISON

We also test the running time of each evaluated method over AR and MIT CBCL. To facilitate the comparison, we report the averaged running time of each layer for multilayer models. For each database, we randomly choose 3, 6, 9 categories to train each model. The averaged runtime based on 20 runs are shown in Figure 7. We can find that (1) the runtime increases with the increasing number of training samples; (2) SRMCF and LGCF usually needs more time than other methods. The main reason may be that it needs more time for the convergence of the algorithm; (3) our VMCF delivers acceptable results in terms of runtime performance. This is mainly due to the gradual decrease in the dimensionality of the basis images.

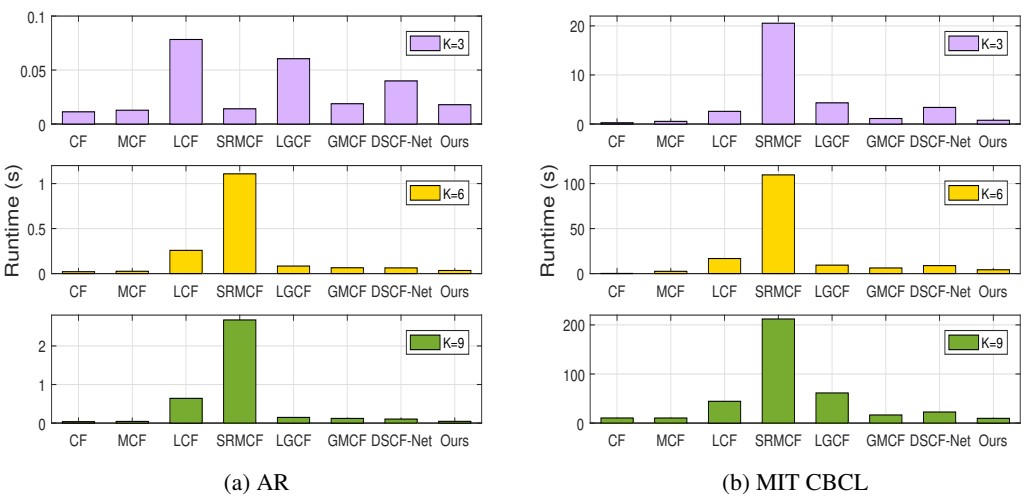

(a) AR                    (b) MIT CBCL

Figure 7: Averaged runtime comparison of each method based on AR and MIT CBCL databases

## 6 CONCLUSION

We mainly discussed the unsupervised locality-preserved representation learning problem for high-dimensional image data. In order to improve the representation and clustering abilities, we technically proposed a new visible multilayer concept factorization method. To capture hidden deep information while slowly reducing data dimensionality, our VMCF designs a multilayer structure equipped with a $D^3$-net in each layer. $D^3$-net jointly incorporates adaptive graph regularized CF, 2D feature extraction for basis images and data reconstruction into a unified framework. In this way, both sample-level and pixel-level neighboring information can be retained simultaneously. Meanwhile, learning low dimensional representations of the basis images layer by layer can not only avoid the loss of valuable information caused by significant dimensionality reduction in one step, but also indirectly improve the quality of reconstructed data and new data representations. We evaluated VMCF for image representation and clustering, and compared the results with related single-layer and multilayer methods. Extensive results demonstrated the effectiveness of our VMCF. In future, more effective multilayer concept factorization strategy will be investigated.

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
