# OpenReview forum: "A novel Visible Multilayer Concept Factorization for Image Data Representation and Clustering"
_ICLR.cc/2025/Conference — ICLR 2025 Conference Withdrawn Submission_

### Official Review · Reviewer_M2W7 · 2024-10-29

**Soundness:** 2
**Presentation:** 2
**Contribution:** 2
**Rating:** 3
**Confidence:** 4

**Summary:**

This paper presents a visible multilayer concept factorization model for image data representation and clustering. The method is called VMCF. VMCF adopts a multilayer framework with decomposition, dimensionality reduction, and data reconstruction in each layer. VMCF introduces the adaptive graph regularized concept learning on the input data of each layer. The experiments are performed on four very old face image datasets.

**Strengths:**

An image data representation learning and reconstruction method is proposed.

**Weaknesses:**

For the paper, some weaknesses are:
1、The introduction of contributions or innovative points is verbose and not prominent.
2、In writing, the representation of data, variables, and symbols is confusing, which increases the difficulty of reading.
3、This article has shortcomings in terms of innovation, research significance, application value, and experimental aspects. In terms of model innovation, it seems that only traditional matrix factorization techniques have been used to obtain representations and reconstruct data, without any innovative models or constraints. In terms of research significance and application value, compared to very mature autoencoders, this work seems to have no advantages. In terms of experiments, it is not sufficient to conduct experiments only on some traditional and simple facial image datasets, and there is no comparison with many deep representation reconstruction models.
4、Lack of complete algorithm pseudocode table.
5、In the method, it seems that all images are factorized one by one. It seems that the method does not use the common information of the whole dataset during learning like 2DPCA.

**Questions:**

see weakness

---

### Official Review · Reviewer_vDnb · 2024-11-01

**Soundness:** 2
**Presentation:** 1
**Contribution:** 2
**Rating:** 3
**Confidence:** 5

**Summary:**

This paper leverages an adaptive graph regularized concept factorization method, which adds an adaptive weight matrix to learn the coefficients in subspace under concept factorization framework. With 2DPCA, it extracts lower dimensional features for deeper layers, obtaining low costs and time.

The multilayer idea seems interesting, but the comparison methods are so old that the advantages are hard to notice. Additionally, writing needs to be improved.

**Strengths:**

The multilayer idea seems interesting. This layer-by-layer strategy decreases the dimension step by step, which can be useful for retaining key information.
The equations are clear.

**Weaknesses:**

The first key weakness of the paper is the comparison methods. And the latest method is DSCF-net, which was published in 2020. It significantly reduces the strengths to the performance and the novelty of this method.
Second, though classic, more recent datasets need to be tested and compared (in the paper the latest is from ETH80, 2003).
Third, no ablation analysis for verifying the ideas.

**Questions:**

1.	As above, new methods and datasets should be compared and tested to support the novelty and performance.
2.	Compare a recent method also based on D3-net, which is the most similar to this method. And discuss the differences.
3.	Ablation analysis is needed. Add or remove the adaptive weight matrix Q. What will the number of layers affect? Also the strategy of the dimension reduction.

---

### Official Review · Reviewer_ms2E · 2024-11-03

**Soundness:** 2
**Presentation:** 2
**Contribution:** 2
**Rating:** 3
**Confidence:** 4

**Summary:**

The paper introduces Visible Multilayer Concept Factorization (VMCF) for image data representation and clustering. VMCF addresses the limitations of traditional Concept Factorization (CF) methods by preserving pixel-level neighborhood information in Two-Dimensional (2D) images, which is often lost in vector-based representations. The proposed method uses a multilayer framework with a 'Decomposition, Dimensionality reduction, and Data reconstruction' network (D$^3$-net) in each layer. VMCF employs adaptive graph regularized concept learning and 2D feature extraction to maintain locality-preserving features and reduce information loss during dimensionality reduction. The method is evaluated on several public image databases, and the results show that VMCF outperforms other state-of-the-art algorithms.

**Strengths:**

- It makes sense to take into account sample-level and pixel-level neighborhood information.


- Experiments are well conducted on public image databases

- The detailed explanation of the VMCF framework, including the adaptive graph regularized concept factorization and the 2D feature extraction process, shows a deep understanding of the subject matter.

**Weaknesses:**

**Novelty**: Though reasonable to consider sample-level and pixel-level information, the methodology in this paper is not trivial, such as the direct use of existing 2DPCA, adaptive graph regularized term similar to [1].

[1] Adaptive graph regularized non-negative matrix factorization with self-weighted learning for data clustering

**Complexity Analysis**: The paper could benefit from a more detailed analysis of the time complexity of the VMCF method, especially when compared to other methods. Though runtime comparison provided in Sec. 5.3, it is expected to show how different methods converge as time goes.

**Scalability Concerns**: The paper does not address the scalability of VMCF to very large datasets or high-dimensional data. It would be useful to know how the method performs as the size and dimensionality of the data increase.

**Comparison with Other Deep Learning Methods**: The paper compares VMCF with traditional and some deep models (the most recent method dates back to 2020), but it would be informative to see how VMCF stacks up against other recent deep learning approaches for image representation and clustering. Some related work is recommended below.

[1] Deep embedded multi-view clustering via jointly learning latent representations. 2022
[2]  Learning deep generative clustering via mutual information maximization. 2022
[3] EDCWRN: efficient deep clustering with the weight of representations and the help of neighbors. 2023

**Typo**: Line 313: Table. 1 --> Table 1, Line 160: unfold, reduce --> unfolds, reduces, etc.

**Questions:**

* How does the computational complexity of VMCF compared to more recent state-of-the-art methods, especially deep learning-based approaches?


* What are the limitations of VMCF in terms of scalability to very large datasets or high-dimensional data?


* How does VMCF perform in terms of convergence speed compared to other methods, particularly in deep models?


* Are there any specific cases or conditions under which VMCF is expected to perform particularly well or poorly?


* Could the authors provide more insights into how the choice of hyperparameters, such as the number of layers and the rank r, affects the performance of VMCF?


* What are the implications of the multilayer structure on the interpretability of the learned representations?

---

### Note · Authors · 2024-11-13

I have read and agree with the venue's withdrawal policy on behalf of myself and my co-authors.